# On the optimal presence strategies for workplace during pandemics: A COVID-19 inspired probabilistic model

Mansoor Davoodi[1]*, Abhishek Senapati[1], Adam Mertel[1], Weronika Schlechte-Welnicz[1], Justin M. Calabrese[1,2,3]

1 Center for Advanced Systems Understanding (CASUS), Helmholtz-Zentrum Dresden Rossendorf (HZDR), Görlitz, Germany, 2 Department of Ecological Modelling, Helmholtz Centre for Environmental Research – UFZ, Leipzig, Germany, 3 Department of Biology, University of Maryland, College Park, MD, United States of America

* m.davoodi-monfared@hzdr.de

## Abstract

During pandemics like COVID-19, both the quality and quantity of services offered by businesses and organizations have been severely impacted. They often have applied a hybrid home office setup to overcome this problem, although in some situations, working from home lowers employee productivity. So, increasing the rate of presence in the office is frequently desired from the manager's standpoint. On the other hand, as the virus spreads through interpersonal contact, the risk of infection increases when workplace occupancy rises. Motivated by this trade-off, in this paper, we model this problem as a bi-objective optimization problem and propose a practical approach to find the trade-off solutions. We present a new probabilistic framework to compute the expected number of infected employees for a setting of the influential parameters, such as the incidence level in the neighborhood of the company, transmission rate of the virus, number of employees, rate of vaccination, testing frequency, and rate of contacts among the employees. The results show a wide range of trade-offs between the expected number of infections and productivity, for example, from 1 to 6 weekly infections in 100 employees and a productivity level of 65% to 85%. This depends on the configuration of influential parameters and the occupancy level. We implement the model and the algorithm and perform several experiments with different settings of the parameters. Moreover, we developed an online application based on the result in this paper which can be used as a recommender for the optimal rate of occupancy in companies/workplaces.

## Introduction

During the pandemics, particularly the very recent coronavirus (SARS-CoV-2, hereafter COVID-19) pandemic, most of the services provided by the companies, organizations and governments' facilities have been affected significantly in terms of quality and quantity [1–6]. On one hand, most of the employees have been at risk of infection and on the other hand, the

**Data Availability Statement:** There is no particular data in this study, and the algorithms are available online on our website, https://www.where2test.de/optimization.

**Funding:** This work was partially funded by the Where2Test project, which is financed by Saxon Ministry for Science, Culture, and Tourism (SMWK) with tax funds on the basis of the budget approved by the Saxon State Parliament. This work was also partially funded by the Center of Advanced Systems Understanding (CASUS) which is financed by Germany's Federal Ministry of Education and Research (BMBF) and by the Saxon Ministry for Science, Culture and Tourism (SMWK) with tax funds on the basis of the budget approved by the Saxon State Parliament. The funders had no role in study design, data collection and analysis, decision to publish, or preparation of the manuscript.

**Competing interests:** The authors have declared that no competing interests exist.

work-flows and functions have become slow. As COVID-19 spreads mostly through direct/close contacts between individuals, the risk of getting infected remains despite adopting some recommended personal hygiene measures in the workplace (e.g., wearing a mask, maintaining distance, etc.). To control and overcome these issues, most of the companies mainly followed three strategies: (*i*) reducing the number of on-site employees by teleworking, (*ii*) screening employees via periodically testing, and (*iii*) vaccinating the employees [7–9]. These strategies have worked quite well to decrease the number of infected cases among the employees, however, the efficiency and the productivity of the employees, particularly who use to work in collaborative environment or where the physical presence of the employee is necessary may get affected while working from home [1, 10]. Therefore, the managers mostly prefer to increase the availability of the employees in the offices to increase work and service productivity. It is very evident that there is a potential trade-off between productivity and the risk of infection, and an important question that arises here is *"what would be the optimal presence rate in the workplace during the pandemic?"*.

During the pandemic, many companies and institutions tried to adapt to the new situation by lowering the possibilities of getting infected at the workplace and keep the employees safe during the work time. Therefore, implementation of effective strategies in the workplace regarding the duration of working hour, presence rate of the employees, level of personal protection for avoiding infection is very important from the management perspective. An ample amount of mathematical models have been developed specifically to investigate several aspects of transmission mechanism and control strategies of COVID-19 [11–15]. In addition, several mathematical and statistical model based investigation have also been carried out to gain better insights on the outcomes of the proposed strategies particularly tailored to different organizations. Most of these studies focused on implementing organizational strategies mainly in the workplaces like hospitals and health care facilities, nursing homes, offices, schools, etc. These studies investigated several aspects like effectiveness of control measures in preventing the disease transmission, scheduling work place arrangements to minimize the consequence of infection, strategies for reopening different activities in organizations, etc.

Regarding effectiveness of control measures and surveillance several attempts have been made by the previous studies with the help of mathematical models to quantify the optimal frequency of testing in healthcare environment [16, 17], nursing home [18], timeliness of contact tracing and importance of molecular in preventing disease transmission among health workers and other high risk groups [19], performance of combinatorial group testing [20, 21], importance of self-isolation and contact tracing measures [22], etc. Also, [23] proposed an optimal testing strategy that essentially minimizes the presence of pre-symptomatic and asymptomatic employees in the workplace.

In the context of scheduling the workplace arrangement, a *desynchronization strategy* has been proposed in which the workers in the healthcare system can be divided into two non-overlapping teams who will be working in alternating weeks to increase the workforce productivity in the facility [24]. Also incorporating the source of infection from inside and outside the hospital, [25] studied the effectiveness of regular testing and *desynchronization* protocol in preventing COVID-19 infection. The effect of cyclic 4-day work and 10-day lockdown strategy in workplace has been investigated by [26]. In nursing home and long-term care facilities, [27] constructed a mathematical model based on bipartite network consists of health care workers and the residents and investigated how the restructured interactions in the bipartite network can change the outcome of the epidemics in the facility by exploiting the strategy of *shied immunity*, where the recovered people increase their interactions with susceptible population to protect them from the infection. In order to support the decision-making process, [28] developed a strategy to predict the risk of infection among the health workers and other

employees in the essential sectors by taking into account the working process, proximity between employees, type of activity, etc.

Several recommendations have been proposed in reopening different activities in workplace. How enhanced level of testing, contact-tracing, adoption of facial mask, home quarantine can play a significant role in relaxing social distancing interventions, which is not always feasible to maintain in the facilities like hospital [29, 30]. It has been recommended that with the adoption of large-scale trace and test intervention along with moderate social distancing, the schools in Île-de-France region can be reopened with maximum attendance of 50% that could avoid second wave [31]. How designing an optimal employee screening strategy can reduce the on-site infection in a workplace and based on this strategy, the feasibility of return to work policy are investigated in [32].

From the above discussions, it is clear that most of the studies have focused on quantifying the impact of implementing different measures in work to reduce infection by developing mathematical and statistical models. Also, some of the studies are devoted to recommending strategies for scheduling workplace arrangements and reopening different activities. However, the aspect of how the influential factors related to preventive measures such as testing frequency, contact rate, and vaccination rate of employees play significant roles in quantifying the optimal presence rate as well as *home-office productivity* factor paid no attention in the previous studies. Motivated by this circumstance, in this paper, we develop a model of disease transmission within companies and organizations that allows us to identify the optimal balance of the mentioned components. To this end, we incorporate influential factors such as the local incidence level, transmission rate of COVID-19, testing frequency, rate of vaccination and rate of contact among the employees. These factors are the most important determinants of the optimal strategy for home-vs-office work. In addition to them, one crucial factor is the work efficiency at home compared to the work efficiency at the office. This factor varies from organization to organization depending on the type of organization and their inputs, processes and outputs. For the sake of simplicity, hereafter, we call the ratio of efficiency at home to efficiency at office by *productivity*.

The proposed model in this paper is based on a probabilistic framework that allows to model spreading of the COVID-19 under a setting of the mentioned influential factors and computes the expected number of infected employees when some infection arrives at the facility. As the advantages of this probabilistic framework, it is simple, fast and practical with the ability to be customized for a specific organization. Moreover, the proposed probabilistic model can be efficiently used for any size of the population. Based on this model, we developed an optimization application to find Pareto optimal solutions for the objectives of maximizing productivity and minimizing infection risk in organizations and companies. The model and the graphical interface of the application are implemented in such a way that it provides the users full ability to reassign all input parameters and renders the results on any change.

This paper is organized into four sections. In the next section, we introduce the influential parameters and state the problem. In addition, we propose the probabilistic analysis and algorithm to compute the risk of infection as well as the Pareto optimal solutions. In the third section, we perform several experiments and test the model and algorithm in deriving Pareto optimal solutions. Finally, we finished the paper in the fourth section by drawing a conclusion and future directions.

## Modeling the office/workplace presence problem and solution approach

In the last two decades, teleworking and home office have rarely been applied for some companies, organizations, corporations, and government facilities (for the sake of simplicity, in the

**Table 1. The parameters and variables used in the proposed model.**

| Notation | Definition |
|---|---|
| $n$ | The total number of employees |
| $n_v$ | The number of vaccinated employees |
| $n_I$ | The number of infected employees |
| $\beta_u$ | The COVID-19 transmission rate for unvaccinated individuals |
| $\beta_v$ | The COVID-19 transmission rate for vaccinated individuals |
| $prod$ | (home) Productivity factor |
| $\tau$ | The average test interval of the employees |
| $\rho$ | Probability of infection arrives at the facility |
| $\kappa$ | The average contact number per employee per day |
| $occup$ | The presence rate at office (say occupancy) |

rest of this paper we use the company to refer to all of them) regarding the type of services they provide. There are comparison studies that discuss the advantages and disadvantages of tele-working [33, 34]. However, since the beginning of the COVID-19 pandemic, most companies have been forced to follow home office partially. This usually has decreased their productivity rate, but significantly reduces the effect of COVID-19 by preventing (or at least reducing) the spreading of the virus in the facilities. Therefore, there is a trade-off between the rate of presence at the offices and the rate of infection at the offices, and it is more challenging for the companies that their employee work at the offices with a higher rate of productivity compared to working at home.

In this section, we formulate this trade-off as a bi-objective optimization problem to find the optimal rate of presence by considering the risk of infection at the facility. It is notable if the productivity rate of employees of some company (because of the type of task and process it provides) is not reduced by following the home-office strategy, clearly, there is no trade-off between the risk of infection in the offices and productivity, and the optimal trivial strategy is minimum occupancy in the offices. Note that, following this strategy, there is still some possibility of infection for the employees working at home, however, there is no propagation of the disease in the office space. Thus, in this paper, we only consider the companies for which the total productivity is increased by the availability of their employees in the offices. It is notable that there will be still some natural risk for the employees working at home, but the companies do not have direct control over this.

Table 1 shows the key parameters and variables used in the model. The exact definition of them and the model's assumptions are explained as follows.

## Parameters, variables, and assumptions of model

- $n$: The total number of employees who regularly work in the facility and the productivity in the facility would be affected if they got infected.

- $n_v$: The number of employees who are (fully) vaccinated with one of the available COVID-19 vaccines. We count partially vaccinated (e.g., one dose) individuals as non-vaccinated

- $n_I$: The expected number of employees who are infected and can spread the disease in the facility. As soon as they do a test or the disease's symptoms appear, they should be in quarantine, so the facility does not count them as an active employee.

- $\beta_u$ and $\beta_v$: The probability of the virus transmitting from one infected individual to one unvaccinated ($\beta_u$) or one vaccinated ($\beta_v$) individual per one potential contact.

- *prod*: The productivity factor is the productivity of working at home compared to working at the office. This factor varies from facility to facility and from employee to employee, based on their mission and the type of services they provide. Managers (or any decision-maker) can determine this factor based on their experience in working under the COVID-19 pandemic and comparing it with normal situations. Precisely, $prod = \frac{Productivity\ of\ working\ from\ Home}{Productivity\ of\ working\ at\ Office}$. So, *prod* = 1 means the manager believes that there is no difference between productivity for working at home and office, and *prod* = 0 means the manager strictly prefers the employees work at the office.

- $\tau$: On average, how often do the employees conduct a test to detect COVID-19? Precisely, it is the average time interval between two sequential tests and significantly helps to detect the infection before it highly spreads in the facility. Since the symptoms of the COVID-19 disease appear within two weeks [35, 36], we assume $\tau$ varies from one day to 14 days (*incubation period*). It is worth mentioning, for the sake of simplicity, in this paper, we assume the tests are perfect and no sensitivity exists. Otherwise, as an advantage of the proposed model, we can easily apply the test's error as a coefficient in computing the expected time to detect the infection.

- $\rho$: The probability that some infections arrive at the facility per day. This can be computed by considering the number of employees at the facility and the incidence level in the neighborhood of the facility. For example, the number of infections for the last 7 days is reported daily (e.g., see https://www.coronavirus.sachsen.de). So, by utilizing it, we can compute such probability. However, to achieve a more realistic value, we apply the ratio of the local vaccination rate to the employees' vaccination rate.

- $\kappa$: The average number of contacts among the employees that have the potential to transmit the disease. We assume the employees and clients mostly wear masks and keep a distance of at least 1.5 meters when they contact each other.

- *occup*: The rate (percentage) of employees present at the office. This is the only decision variable of the model which should be determined optimally.

As mentioned, all the variables and parameters are given except the decision variable *occup* which is the output of the model. Indeed, the employees are divided into three groups: (*i*) who work at the office, (*ii*) who work from home, and (*iii*) who are infected. The productivity factor of the first group is one, and that of the second and the third groups are *prod* and zero, respectively. So, $n - n_I$ is the number of healthy and active employees and out of them *occup* percentage of employees who works at the offices with full productivity equals one, and the remaining 1 − *occup* percentage of them work from home with productivity *prod*. Therefore, the total productivity of a facility can be determined by the following formula. Note that it is possible to customize this formula and define it based on the input/fellows/output in a company. In this paper, we skip such details and only focus on this basic and general definition of total productivity.

$$Total\ Productivity = occup \times (n - \mathbb{E}(n_I)) + prod \times (1 - occup) \times (n - \mathbb{E}(n_I)) \qquad (1)$$

   Therefore, one objective function of the model is **maximizing the total productivity**, and the other one is **minimizing the expected number of infections**, $\mathbb{E}(n_I)$. There is a clear conflict between these two objectives. Note that, *prod* is the productivity factor determined by the decision-maker. Without loss of generality, we assume it is between zero and one, and if $prod \geq 1$, there is a trivial optimal solution *occup* = 0, that is working from home strictly is preferred to working at the office, and it results in minimizing the number of infected employees.

The most important part of the objective functions is computing the expected number of infected employees, $\mathbb{E}(n_I)$, by arriving some infections at the facility. Indeed, we need to compute it as a function over time and based on the influential parameters mentioned in Table 1 which are explained in detail in the next subsections.

### Probabilistic framework for computing the number of infected employees

We start by a full contact network of unvaccinated employees and present the approach. Then we extend it to two groups of vaccinated and unvaccinated employees. Employees can contact each other with the same probability, however, the average number of contacts per day is bounded by $\kappa$. We suppose an infection arrives at the facility at the time (day) $t = 0$, and compute the probability of an arbitrary individual being infected after $\Delta t$ days when the transmission rate of the virus is $\beta_u$ and the number of contacts is $\kappa$ per day. Let denote this value with the probability function $P_I(\Delta t, n, \beta_u, \kappa)$. By having this probability function, the expected number of infected individuals can be easily computed by

$$\mathbb{E}[(n_I(\Delta t, n, \beta_u, \kappa)] = 1 + (n-1) \times P_I(\Delta t, n, \beta_u, \kappa). \tag{2}$$

For the sake of simplicity, in the rest of the paper, let denote the probability function $P_I(\Delta t, n, \beta_u, \kappa)$ by $P_I(\Delta t)$ for known variables of $n$ and $\kappa$. Thus, computing the probability of infection per employee results in achieving the expected number of infected employees. Let denote the source of infection at $t = 0$ by $s$. Let $u$ be an employee who stays healthy till $\Delta t - 1$. There are two ways for infecting $u$ at day $\Delta t$; via some direct contact with $s$, or via contact with one of the $n - 2$ other employees. The transmission probability for each of the contacts is $\beta_u$, and the probability of infection for the source is one, while the probability of infection for the other employees is $P_I(\Delta t - 1)$. See Fig 1. That means the probability of an arbitrary employee like $u$ staying healthy after $c$ contacts with $s$ is $(1 - 1 \times \beta_u)^c$. While the probability of $u$ stays healthy

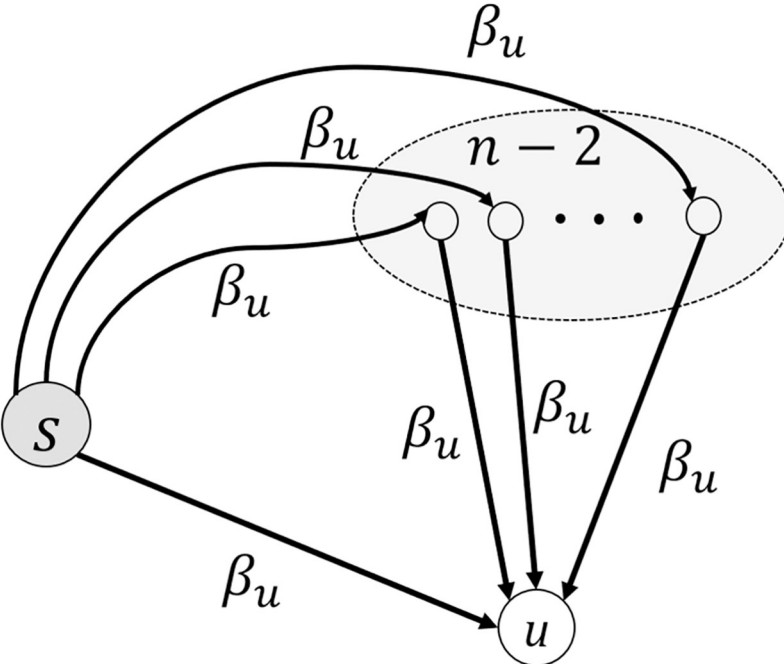

**Fig 1. Infection possibilities of $n$ employees which one of them ($s$) is infected.**

after $c$ contacts with the other $n-2$ employees in day $\Delta t$ is $(1 - P_I(\Delta t - 1) \times \beta_u)^c$. Therefore, the infection probability can be presented as a recursive function. Since we assume $\kappa$ contacts for each employee per day and all of them have an equal chance to occur, this function can be written as

$$P_I(\Delta t) = 1 - (1 - P_I(\Delta t - 1)) \times \left[ (1 - \beta_u)^{\kappa \frac{1}{n-1}} \times (1 - P_I(\Delta t - 1)\beta_u)^{\kappa\left(1 - \frac{1}{n-1}\right)} \right], \qquad (3)$$

where $\kappa \frac{1}{n-1}$ is the expected number of contacts between employee $u$ and the infected source employee $s$, and $\kappa\left(1 - \frac{1}{n-1}\right)$ is the expected number of contacts between employee $u$ and the other employees except $s$. Note that, $(1 - P_I(\Delta t - 1))$ is the probability of the employee staying healthy till day $\Delta t - 1$.

Eq 3 computes the probability of an arbitrary employee getting infected after $t = \Delta t$ days when the employees have $\kappa$ contacts per day and present at the office with $occup = 1$, that is, full occupancy. For $0 \leq occup \leq 1$ as the rate of presence at the office, all the employees (include $s$) stay (healthy) at home with the expected probability $1 - occup$. So the probability of having one contact between $u$ and $s$ is $occup^2$. Therefore, the infection probability formula for known $n$, $\kappa$ and $occup$ can be extended as follows

$$P_I(\Delta t) = 1 - (1 - P_I(\Delta t - 1))\left[ (1 - \beta_u)^{occup^2 \times \kappa \frac{1}{n-1}} \times (1 - P_I(\Delta t - 1)\beta_u)^{occup \times \kappa\left(1 - \frac{1}{n-1}\right)} \right]. \quad (4)$$

Also, we set $P_I(0) = 0$ as the scenario supposed here. After computing the probability of infection, the expected number of infected employees can be computed by applying Eq 2 easily.

Now, let's extend the above computation to two different groups, say vaccinated and unvaccinated employees. We consider a probability of the disease transmission $\beta_v$ ($\beta_v \ll \beta_u$) for the vaccinated group. Without loss of generality, we consider the source of infection $s$ as an unvaccinated employee and compute the probability of infection in two cases ($i$) for an unvaccinated employee $u$, and ($ii$) for a vaccinated employee $v$ (see Fig 2). There are three ways for infecting $u$; via direct contact with $s$, via contacting with one of the $n_v$ vaccinated employees, and via contact with one of the $n - n_v - 2$ unvaccinated employees. The transmission probability for each of such contacts is $\beta_u$ because $u$ is an unvaccinated employee, however, the transmission probability for the vaccinated group is $\beta_v$. Let $P_I^u(\Delta t - 1)$ and $P_I^v(\Delta t - 1)$ be the probability of infection for an unvaccinated employee and a vaccinated employee, respectively. Thus, the

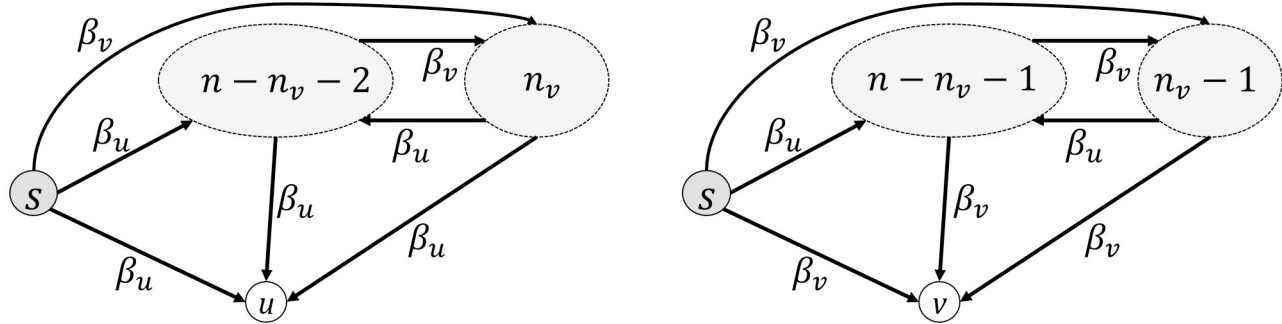

**Fig 2. Infection possibilities of $n$ employees which one of them ($s$) is infected and there are two groups of vaccinated and unvaccinated employees.**
The left subfigure shows the scenario for an unvaccinated employee $u$ and the right one shows the scenario for a vaccinated employee $v$.

recursive equation for them can be written below

$$
\begin{aligned}
P_I^u(\Delta t) = 1 - (1 - P_I^u(\Delta t - 1)) \Big[ & (1 - \beta_u)^{occup^2 \times \kappa \frac{1}{n-1}} \\
& \times (1 - P_I^u(\Delta t - 1)\beta_u)^{occup \times \frac{n-n_v-1}{n-1} \times \kappa \left(1 - \frac{1}{n-1}\right)} \\
& \times (1 - P_I^v(\Delta t - 1)\beta_v)^{occup \times \frac{n_v}{n-1} \times \kappa \left(1 - \frac{1}{n-1}\right)} \Big],
\end{aligned}
\tag{5}
$$

and

$$
\begin{aligned}
P_I^v(\Delta t) = 1 - (1 - P_I^v(\Delta t - 1)) \Big[ & (1 - \beta_v)^{occup^2 \times \kappa \frac{1}{n-1}} \\
& \times (1 - P_I^u(\Delta t - 1)\beta_v)^{occup \times \frac{n-n_v-1}{n-1} \times \kappa \left(1 - \frac{1}{n-1}\right)} \\
& \times (1 - P_I^v(\Delta t - 1)\beta_v)^{occup \times \frac{n_v}{n-1} \times \kappa \left(1 - \frac{1}{n-1}\right)} \Big],
\end{aligned}
\tag{6}
$$

where $P_I^u(0) = P_I^v(0) = 0$. One advantage of these recursive formulas is that $P_I^u(\Delta t)$ and $P_I^v(\Delta t)$ can be efficiently computed in linear time to $\Delta t$ using a simple bottom-up approach. Finally, the expected number of infected employees after $\Delta t$ days of introducing an infection can be computed as follows

$$
\mathbb{E}[n_I(\Delta t, n, n_v, \beta_u, \beta_v, \kappa, occup)] = 1 + (n - n_v - 1) \times P_I^u(\Delta t) + n_v \times P_I^v(\Delta t).
\tag{7}
$$

Thus, Eq 7 provides a recursive formula to compute the number of infections over time. An implementation of this equation and comparison with simulation results is presented in the Appendix.

## Computing optimal presence rate

As explained, the expected number of infected employees after arriving some infection to the facility can be computed using Eq 7 as a function over time and based on the mentioned influential parameters. For the sake of simplicity, we denote this expected number by $\mathbb{E}[n_I(\Delta t)]$. Let $\rho$ be the probability of infection arriving at the facility. This probability can be determined using two straightforward approaches. First, using the (recent) historical data of incidence in the company, and second, using the local incidences and applying the ratio of the vaccination rate in the neighborhood of the facility to the vaccination rate in the company. Therefore, the cumulative number of infected employees for a time interval of $T$ days can be computed as follows

$$
\mathbb{E}(n_I) = Accum_I(T, \rho) = \rho \times Z(T),
\tag{8}
$$

where $Z(T)$ is defined as below

$$
Z(T) = \begin{cases}
\mathbb{E}[n_I(0)] + \mathbb{E}[n_I(1)] - \dfrac{\mathbb{E}[n_I(1)] \times \mathbb{E}[n_I(0)]}{n} & , \text{ if } \quad T = 1 \\[3mm]
\mathbb{E}[n_I(T)] + Z(T - 1) - \dfrac{\mathbb{E}[n_I(1)] \times Z(T - 1)}{n} & , \text{ if } \quad T \geq 2
\end{cases}
\tag{9}
$$

Note that $Z(T)$ is a linear recursive formula to compute the cumulative number of infections for $t = 0, 1, \ldots, T$ by removing the expected overlapped infected employees over time.

Eq 8 computes the number of all infected in a time interval $T$ days where the probability of arriving an infection is $\rho$ per day. If the employees do a test every $\tau$ days on average, and they

uniformly distribute in the test interval (having $k = \frac{n}{\tau}$ tests per day on average), the expected days to detect the infection can be computed using the following equation

$$\bar{\tau} = [1 - (1 - Pr(t))^k] + \sum_{t=2}^{\tau} \left[ t \times (1 - (1 - Pr(t))^k) \times \prod_{t'=1}^{t-1} (1 - Pr(t'))^k \right], \quad (10)$$

where $Pr(t) = \frac{\mathbb{E}[n_I(t)]}{n}$ is the probability of infection per employee after $t$ days of arriving some infection, so, $1 - (1 - Pr(t))^k$ is the probability of detecting at least one infected employee by testing a subgroup of $k$ after $t$ days an infection arrives the facility. For detecting the infection after $t > 1$ days of arriving, we need to compute the probability of it has been not detected in days before $t$, i.e. $t' < t$, which is computed using $\prod_{t'=1}^{t-1} (1 - Pr(t'))^k$.

$\bar{\tau}$ in Eq 10 shows the expected time to detect an infection if the employees do a test every $\tau$ days on average. Therefore, the expected number of infected employees before detecting an infection is obtained by $\mathbb{E}[n_I(\tau, n, n_v, \beta_u, \beta_v, \kappa, occup)]$. So, the total productivity (see Eq 1) of a facility can be determined for any given $occup$ in $[0, 1]$, and based on the other introduced influential parameters.

## A practical multi-objective solution approach

As described before, the optimal office presence strategy is formulated in the framework of a bi-objective optimization problem as follows

$$
\begin{aligned}
&Minimize \quad \mathbb{E}(n_I), \\
&Maximize \quad Total\ Productivity, \\
&\quad Subject\ to: \\
&\quad occup \geq An\ Occupancy\ Threshold.
\end{aligned}
\quad (11)
$$

The first objective is minimizing the expected number of infected employees presented in Eq (8), and the second objective is maximizing the total productivity of the company presented in Eq (1). The only constraint which we considered in the model is related to minimum possible occupancy in the companies, *Occupancy Threshold*. That is a limit on the minimum number of employees that have to be present at the facility in order to process some tasks for which the physical presence of the employees is required. Further, it is possible to add more lower- or upper-bound constraints on the occupancy variable. The model has two conflicting objectives. Clearly, by increasing the occupancy in a company, the expected number of infected employees will increase as well. However, this is not the case for the second objective, because by increasing the occupancy, the number of infected employees will also increase, and consequently, since the productivity of the infected employees is zero, the total productivity will decrease. Fig 3 shows the effect of occupancy on the first and second objective separately for two different scenarios of the test interval, one test per week (blue curve) and one test per two weeks (red curve). As expected, the number of infections is a strictly increasing function of the rate of presence, while, the productivity function is increasing for the low rate of presence and after reaching the maximum productivity (e.g., for $occup \approx 88\%$ in the first scenario and for $occup \approx 65\%$ in the second one) it decreases by increasing presence rate.

The outcome of the optimization problem presented in model 11 is a set of trade-off solutions– called Pareto optimal– a solution that is not improved for an objective unless it sacrifices the other objective [37]. There are different approaches to solving multi-objective optimization problems, such as the weighted-sum method [38], the lexicographic method [39], the $\epsilon - constraint$ approach [40], the goal programming [41], evolutionary algorithms

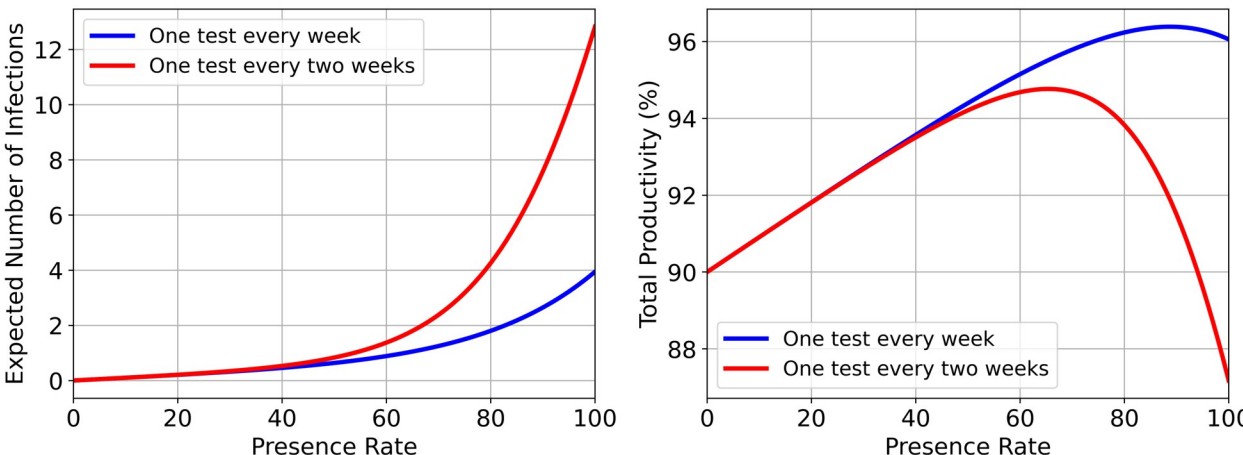

**Fig 3. Effect of occupancy on the expected number of infections (left panel) and on the total productivity (right panel) for two intervals, one test per week and one test per two weeks.** In these simulations, we assumed a company with 100 employees which half of them are vaccinated, with 15% contact rate, and the home productivity rate 0.90.

[37, 42], etc. All of these approaches have their own advantages and disadvantages. Some of them have high time complexity, some of them work only for differentiable functions or convex models, some of them have no guarantee to find all the Pareto-optimal solutions. Thus, we propose a practical quick method to compute all Pareto optimal solutions of the model 11 efficiently.

The method utilizes some observations to solve the model. The, *occup* is a continuous variable, but in reality, there are at most $n$ possible choices for the number of employees present in the facility. So, at first, for any $occup \in \{0, \frac{1}{n}, \frac{2}{n}, \ldots, \frac{n}{n}\}$ which satisfy the occupancy threshold, we compute the expected detection time based on Eq (10), and the expected number of infected employees and its corresponding total productivity based on Eqs (1) and (7), respectively. All of these steps can be handled in $O(n\tau)$ time using the presented linear recursive equations. Let denote this solution set by $S$. Now, we can find the *non-dominated* solutions of $S$ and report them as the Pareto-optimal optimal solutions of the problem. A solution $\bar{s} \in S$ is a non-dominated solution, if there is no other solution in $S$ such that it is better than $\bar{s}$ in both objectives, i.e., its expected number of infections is less than the expected number of infections of $\bar{s}$, and simultaneously, its productivity is more than the productivity of $\bar{s}$. Since the solutions of $S$ are constructed one by one, from the minimum expected infections to the maximum one, the Pareto-optimal solutions of the problem can be computed in linear time using a *sweep-line* approach [43]. The pseudocode of the proposed algorithm is presented as follows.

**Algorithm 1** Computing Pareto Optimal Office Presence Strategies

```
Input: Company's Parameters (the notation mentioned in Table 1)
Output: All Pareto Optimal Strategies
max_prod ← −1
i ← 0
while i ≤ n do
  occup ← i/n
  if occup ≥ Occupancy_Threshold then
    Compute τ̄ using Eq (8)
    Compute expected number of infections using Eq (7)
    Compute the total productivity using Eq (1) and denote it by TP
    if TP > max_prod then
      Report the current strategy as a Pareto optimal strategy
```

```
      max_prod ← TP
    end if
  end if
  i ← i + 1
end while
```

## Simulation results and discussions

In this section, we show some implementation results of the proposed model and method for computing Pareto optimal solutions of the objectives, minimizing the expected number of infections and maximizing productivity. Since there are several influential input parameters in the proposed model, there are so many possible combinations of them as well. Therefore, we shortly display some outputs of the model, and instead, we provided an online optimization tool based on the model to give any possible combinations of the influential parameters as the input and observe the output. The basic version of this tool is available at www.where2test.de.

To run the simulations, we consider a middle-size company with $n = 100$ employees and suppose the 7-day incidence rate per 100,000 population is 500 individuals. Thus, the average probability of infection per employee in a week will be $\frac{500}{100,000}$, and consequently, the probability of infection arrives at the company is $\rho = 1 - \left(1 - \frac{500}{100000}\right)^n$. In this study, we specifically consider two scenarios based on the disease transmission rate for unvaccinated employees ($\beta_u$). First, we choose the baseline value as $\beta_u = 0.04$ from a possible range of values reported in the previous study [44]. This baseline scenario can be regarded as the situation where the different variants of COVID-19 like *Alpha* and *Delta* are the dominant variants and cause infections. On the other hand, the growing evidences [45–47] suggest that the recently identified *Omicron* variant is more transmissible than the previous ones. Therefore, as the second choice, we consider the situation where the transmissibility increases 2.5-folds, i.e. $\beta_u = 0.1$. Also, the transmission rate of vaccinated employees to $\beta_v = (1 - 0.80)\beta_u$. That means 80% immunity for vaccinated individuals is assumed which is in accordance with the efficacy of the available COVID-19 vaccines [48–51].

Further, we assume two different values for the other input parameters. The home productivity factors, *prod* = 0.6 and *prod* = 0.9, and two vaccination rates of employees 0.4 and 0.8. For the test interval, $\tau$, we consider the simulations for two test intervals, one test every week (i.e., $\tau = 7$ days), and one test every two weeks (i.e., $\tau = 14$ days), and for the number of contacts, $\kappa$, we consider *low* and *high* rate of contacts. The low-case scenario assumes every employee who presented at the office has $\kappa = 5 + 0.10 \times occup \times n$ contacts with the other employees, where *occup* is the rate of presence at the offices. In fact, we assume 5 contacts (can be considered as unavoidable contacts) per employee as a constant number, plus 10 percent of the employees who are present at the offices. Likewise, the high rate scenario assumes the number of contacts is $\kappa = 5 + 0.20 \times occup \times n$. Note that, this is just a sample setting of the influential parameters and any other setting may replace regarding the availability of the information. Fig 4 displays the result of 15 scenarios for a different settings of input parameters, whereas Table 2 shows explains these 15 input settings.

Each subfigure in Fig 4 shows Pareto optimal solutions and their objective values. The horizontal axis shows the possible occupancy values. The blue diagram and the left vertical axis illustrate the obtained total productivity, and the red diagram and the right vertical axis illustrate the expected number of infected employees per week. Note that, the Pareto optimal solutions are displayed only for the occupancy values which result in such solutions, and the dominated solutions are not displayed. These results together provide useful information for a decision-maker who may consider a threshold as a maximum risk of infection and try to find

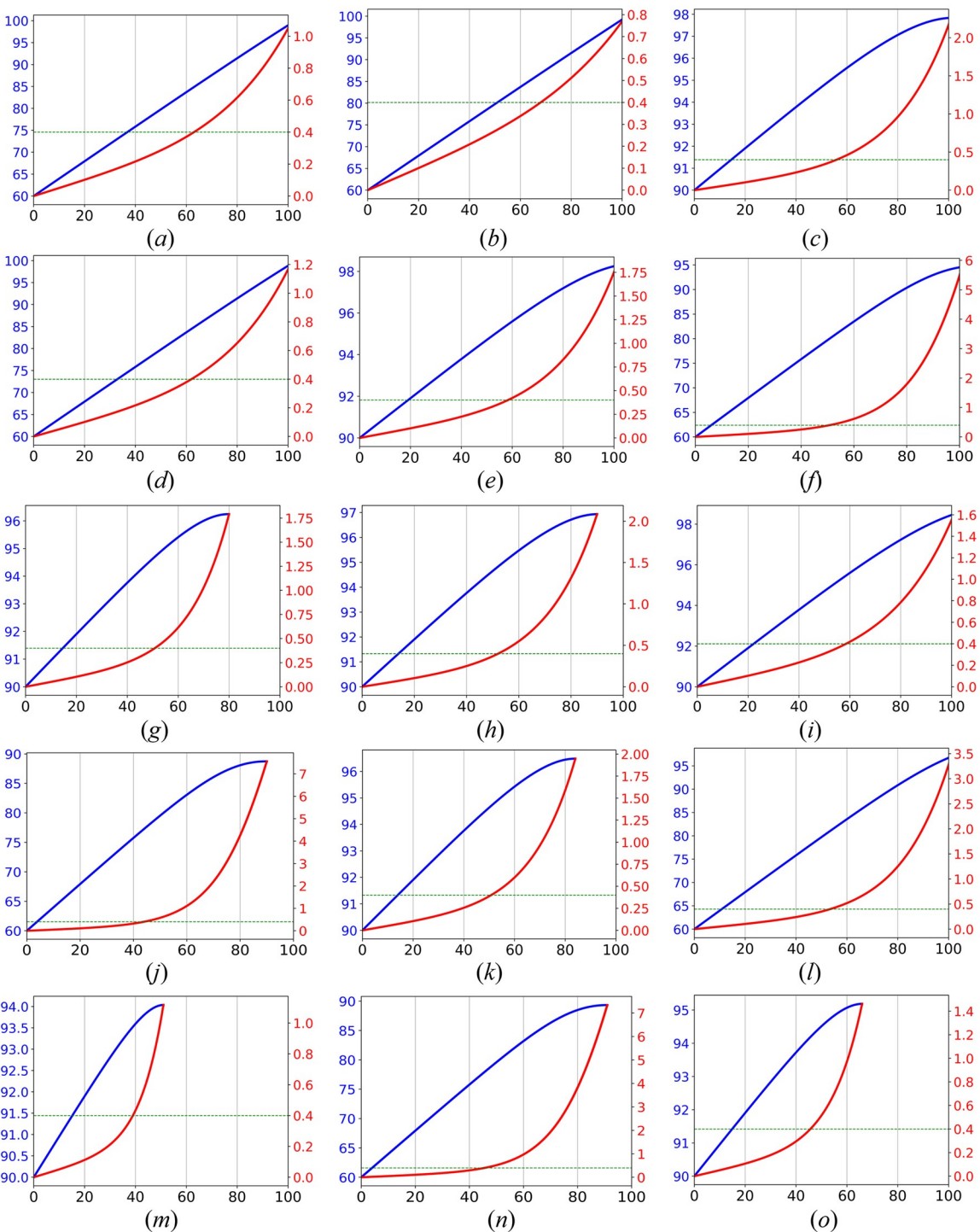

**Fig 4. Pareto optimal solutions for 15 different settings of input parameters.** See Table 2. In each subfigure, the horizontal axis shows the possible occupancy, the blue diagram and the left vertical axis illustrate the obtained total productivity, and the red diagram and the right vertical axis illustrate the expected number of infected employees per week.

**Table 2. 15 different settings of input parameters.** The corresponding Pareto optimal solutions are depicted in Fig 4.

| Fig | $\tau$ | *prod* | vaccine rate | contact rate | $\beta_u$ |
|---|---|---|---|---|---|
| Fig 4(a) | 7 | 0.6 | 0.5 | Low | 0.04 |
| Fig 4(b) | 7 | 0.6 | 0.8 | Low | 0.04 |
| Fig 4(c) | 14 | 0.9 | 0.5 | Low | 0.04 |
| Fig 4(d) | 14 | 0.6 | 0.8 | Low | 0.04 |
| Fig 4(e) | 7 | 0.9 | 0.5 | High | 0.04 |
| Fig 4(f) | 14 | 0.6 | 0.5 | High | 0.04 |
| Fig 4(g) | 14 | 0.9 | 0.5 | High | 0.04 |
| Fig 4(h) | 7 | 0.9 | 0.5 | Low | 0.1 |
| Fig 4(i) | 7 | 0.9 | 0.8 | Low | 0.1 |
| Fig 4(j) | 14 | 0.6 | 0.5 | Low | 0.1 |
| Fig 4(k) | 14 | 0.9 | 0.8 | Low | 0.1 |
| Fig 4(l) | 7 | 0.6 | 0.8 | High | 0.1 |
| Fig 4(m) | 14 | 0.9 | 0.5 | High | 0.1 |
| Fig 4(n) | 14 | 0.6 | 0.8 | High | 0.1 |
| Fig 4(o) | 14 | 0.9 | 0.8 | High | 0.1 |

the maximum productivity by some changes in the vaccination rate of the employees, test interval, or even their number of contacts.

For example, when the 7-day incidences are 500 individuals of 100,000 population (this is almost the average number of incidences in Saxony, Germany from January first till the middle of February 2022, e.g., see https://www.where2test.de/saxony), it means the natural risk (or say the *background risk*) of infection per employee is $\frac{5}{1000}$ per week, and for the whole of the company it is $1 - \left(1 - \frac{500}{100000}\right)^{100} \approx 0.4$. The background risk can be interpreted as the risk of infection if the employees do not come to the office and stay at home for work and maintain the normal social contacts by e.g., going to restaurants and shopping. While the number of incidences is not so high or so low, the decision-makers and managers can use it as a mental guidance like a reference point in determining a proper threshold for the risk of infection in the company. The horizontal green line in each of the subfigure displays such background risk.

For example, if a decision-maker (say a manager in a company) would like to follow a presence strategy whose result never exceeds the natural risk, he/she can consider the occupancy corresponding to the intersection point between the background risk and the computed infection risk, i.e., the green dashed line and the red diagram. For instance, such occupancy in the first scenario in Fig 4(a) ($\tau = 7$, *prod* = 0.6, *vaccination rate* = 0.5, *Low contact rate* and $\beta_u =$ 0.04) corresponds with occupancy 64%, and it will result in 85% of productivity, while in the second scenario, Fig 4(o) ($\tau = 14$, *prod* = 0.9, *vaccination rate* = 0.8, *High contact rate* and $\beta_u =$ 0.10), it is 46% with more than 94% of productivity.

In a reverse usage of the Pareto optimal solutions, a decision-maker may be interested to know what is the risk of infection (e.g., compared to the natural risk) if he/she would like to achieve a particular level of productivity in the company. For instance, in the second scenario, Fig 4(b) ($\tau = 7$, *prod* = 0.6, *vaccination rate* = 0.8, *Low contact rate* and $\beta_u = 0.04$), a level of 70% of the productivity can be achieved with only half of the background risk, i.e., 0.2, while only 25% of employees presence at offices. To compute this, we first find the intersection point of productivity 70% with the productivity diagram (the blue diagram). In the second scenario, it is almost 25% occupancy. Then find the infection risk on the red diagram which is corresponding with such occupancy, which is almost 0.2.

Finally, decision-makers can use several diagrams together to find a sense of the amount of obtained productivity or infection risk by changing some influential parameters, such as test interval and/or vaccination rate among the employees. For instance, if the current situation of a company is similar to the 13th scenario (see Fig 4(*l*)), and occupancy 54% will result in a risk of infection is the same as the background risk and the productivity 81%. Now, if the test interval among the employees rises to $\tau = 14$ days, (see 15-th scenario, Fig 4(*l*)), and the decision-maker still would like to the risk of infection in the company does not exceed the natural risk, he/she have to decrease occupancy to almost 46% and so achieve a productivity level of 78%. A similar analysis can be extracted for investigating the effect of other input parameters. As aforementioned, because of the so many possible combinations of the influential input parameters, there is no way to discuss all sensitivity analyses and observe the changes in the objectives by changing the parameters. Instead, we suggest using the provided online application for this purpose.

## Conclusion and future directions

In this study, we developed a model to compute the optimal trade-off solutions for reducing the risk of infection and increasing the total productivity in companies during the pandemics such as COVID-19. By including fundamentally important parameters such as the local incidence level, the number of contacts among the employees, their average test interval, and vaccination rate, we proposed a probabilistic analysis approach to compute the expected number of infected employees over the course of time. This probabilistic approach can be used to compute the number of infected individuals and probability of infection over time for different groups of people in terms of the probability of infection and spreading the virus.

We assumed two groups of employees with different infection probabilities; vaccinated and unvaccinated, and presented a practical approach to compute the Pareto optimal presence rate of employees to reach the maximum productivity and minimum infected risk. In addition to the incidence level, three critical parameters to obtain the maximum productivity are the home productivity rate of the employees, the contact rate between the employees and the average test interval among the employees. We designed the model as simple as possible to be able to cover and interpret the effect of all influential parameters. The presented approach is linear in terms of time complexity, and it can be simply extended to consider the sensitivity of COVID-19's tests.

The manager of the business and organization can benefit from the outcome of the model in decision processes such as self-testing regulations of the employees and rate of remote working. The Pareto-optimal solutions illustrated a trade-off range between infection risk and productivity, which considerable managers in their decision-making. Further, the model can be extended to more than two groups of employees, e.g, different age groups. A basic implementation of this model and algorithms are available online at https://test.where2test.de. Future extensions of this study may consider the following subjects

- In addition to employees, visitors (i.e., clients and customers) who have direct contact with the employees can be considered for computing the probability of an infection arriving at the facility.

- Heterogeneous groups of employees with different contact rates, physical networks, productivity factors and test frequencies can be considered.

- We defined the total productivity of the company based on individual productivity, that was, the summation of all employees' productivity. However, it is possible for some companies the total productivity is defined based on the type of fellows and tasks in different sections of

the company. To this end, the physical graph of contacts and more details on the outcomes may need as given.

## Appendix: Evaluation of the proposed probabilistic approach

However, we explained the theory behind the probabilistic approach, in this section, we implement the proposed probabilistic approach for computing the expected number of infected employees as well. To this end, we assume different settings of input parameters, and compute the expected number of infections along the time. Indeed, we suppose one infection at time $t = 0$, and compute the expected infections for one month, i.e., $t < 30$ using Eq 7. Fig 5 illustrates the results for different values of $\beta_u$, $\kappa$, $occup$ and $n_v$. In these results, we assumed $n = 150$ employees and $\beta_v = 0.15\beta_u$. In each subfigure, three cases of transmission rates, $\beta_u = 0.05$, $\beta_u = 0.10$ and $\beta_u = 0.15$, are shown. Also, for comparison, we show simulation results (the dots), that is, simulating the companies using a set of $n = 150$ agents (employees) based on the

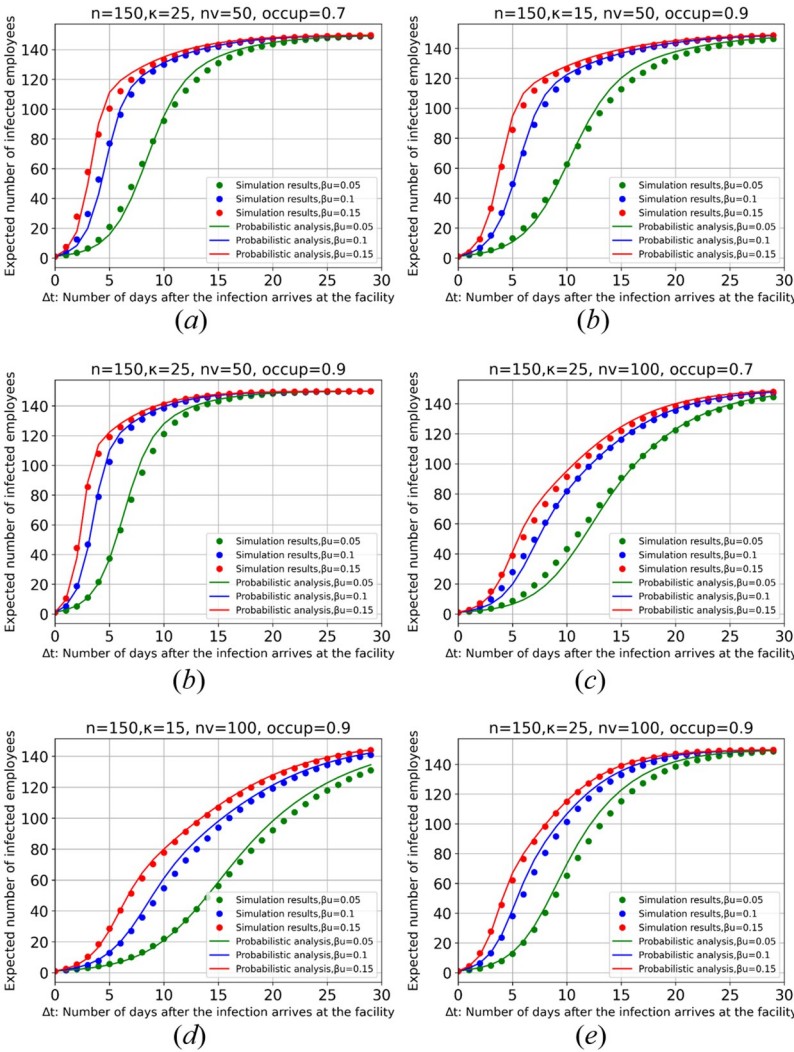

**Fig 5. The simulation results and the probabilistic analysis of the expected number of infected employees for a facility with $n = 150$ employees along time, $\Delta t = 1, 2, \ldots, 29$ days, after the infection arrives.** Six different settings of the influential parameters are shown. The computations are based on Eq 7.

**Table 3. Mean absolute percentage error between the simulation results (average of 100 iterations) and the proposed probabilistic analysis for estimating the number of infected employees for six different settings of the influential parameters.**

|  | $\beta_u = 0.05$ | $\beta_u = 0.10$ | $\beta_u = 0.15$ |
|---|---|---|---|
| Fig 5(a) | 0.055 | 0.042 | 0.041 |
| Fig 5(b) | 0.057 | 0.024 | 0.026 |
| Fig 5(c) | 0.028 | 0.017 | 0.014 |
| Fig 5(d) | 0.078 | 0.057 | 0.040 |
| Fig 5(e) | 0.051 | 0.053 | 0.024 |
| Fig 5(f) | 0.064 | 0.036 | 0.014 |
| **Average** | **0.056** | **0.038** | **0.027** |

parameter's setting. To reduce the effect of random number generators, we ran the simulations for 100 times and report the average number of infected agents. Table 3 shows the *mean absolute percentage error* for each figure and the average of them. As it can be seen, the average (the last row of the Table) percentage of difference between the results is 5.6%, 3.8% and 2.7% for transmission rates $\beta_u = 0.05$, $\beta_u = 0.10$ and $\beta_u = 0.15$, respectively. The results show the probabilistic analysis estimates the number of infected individuals with high accuracy, while it is fast and flexible approach to apply for heterogeneous group of people with different transmission rate and number of contacts.

## Supporting information

**S1 Text.**
(TXT)

## Author Contributions

**Formal analysis:** Mansoor Davoodi.

**Investigation:** Mansoor Davoodi.

**Methodology:** Mansoor Davoodi.

**Project administration:** Justin M. Calabrese.

**Software:** Adam Mertel.

**Supervision:** Mansoor Davoodi, Justin M. Calabrese.

**Validation:** Mansoor Davoodi.

**Visualization:** Mansoor Davoodi.

**Writing – original draft:** Mansoor Davoodi, Abhishek Senapati, Adam Mertel, Weronika Schlechte-Welnicz, Justin M. Calabrese.

**Writing – review & editing:** Mansoor Davoodi, Abhishek Senapati, Adam Mertel, Weronika Schlechte-Welnicz, Justin M. Calabrese.

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
