## [Decision Letter · Decision Letter 0]

27 Jan 2023

PONE-D-23-00247On the optimal presence strategies for workplace during pandemics: A COVID-19 inspired probabilistic modelPLOS ONE

Dear Dr. DavoodiMonfared,

Thank you for submitting your manuscript to PLOS ONE. After careful consideration, we feel that it has merit but does not fully meet PLOS ONE’s publication criteria as it currently stands. Therefore, we invite you to submit a revised version of the manuscript that addresses the points raised during the review process.

We look forward to receiving your revised manuscript.

Kind regards,

Muhammad Farhan Bashir

Academic Editor

PLOS ONE

“This work was partially funded by the Center of Advanced Systems Understanding (CASUS), which is financed by Germany’s Federal Ministry of Education and Research (BMBF) and by the Saxon Ministry for Science, Culture, and Tourism (SMWK) with tax funds on the basis of the budget approved by the Saxon State Parliament.”

“This work was partially funded by the Where2Test project, which is financed by SMWK

with tax funds on the basis of the budget approved by the Saxon State Parliament.

This work was also partially funded by the Center of Advanced Systems Understanding

(CASUS) which is financed by Germany’s Federal Ministry of Education and Research

(BMBF) and by the Saxon Ministry for Science, Culture and Tourism (SMWK) with

tax funds on the basis of the budget approved by the Saxon State Parliament.”

“This work was partially funded by the Center of Advanced Systems Understanding (CASUS), which is financed by Germany’s Federal Ministry of Education and Research (BMBF) and by the Saxon Ministry for Science, Culture, and Tourism (SMWK) with tax funds on the basis of the budget approved by the Saxon State Parliament.”

“NO authors have competing interests.”

Additional Editor Comments:

Dear Authors,

As you can see that reviewers have raised some serious issues to be resolved in your submission. Kindly pay close attention to these comments and submit revised manuscript timely.

Reviewers' comments:

Reviewer's Responses to Questions

**Comments to the Author**

1. Is the manuscript technically sound, and do the data support the conclusions?

Reviewer #1: Partly

Reviewer #2: Yes

Reviewer #3: Yes

2. Has the statistical analysis been performed appropriately and rigorously? 

Reviewer #1: Yes

Reviewer #2: Yes

Reviewer #3: Yes

3. Have the authors made all data underlying the findings in their manuscript fully available?

Reviewer #1: Yes

Reviewer #2: Yes

Reviewer #3: Yes

4. Is the manuscript presented in an intelligible fashion and written in standard English?

Reviewer #1: No

Reviewer #2: Yes

Reviewer #3: Yes

5. Review Comments to the Author

Reviewer #1: many thanks to the editors for the invitation. I have read your work carefully.. Specific comments are as follows.

-The abstract should briefly describe the research background and policy implications. Please add a graphical abstract

- Research gaps should be well mentioned in the introduction. A good research gap can give the reader more insight.

-how can the governments benefit economically from your research ?

-Theoretical analysis needs to reflect the authors' deeper thinking. Please improve it further and discuss it with the innovation of the paper.

-why the method you have used is better than other methods ? how did you improve it?

-Some of the most recent literature (last three years) should be considered and updated.

The following papers can be good examples to help you improve your paper to be a case can be applied worldwide:

-Dagestani, A.A.; Qing, L.; Abou Houran, M. What Remains Unsolved in Sub-African Environmental Exposure Information Disclosure: A Review. J. Risk Financial Manag. 2022, 15, 487. https://doi.org/10.3390/jrfm15100487

-Dagestani, A. A. (2022). An Analysis of the Impacts of COVID-19 and Freight Cost on Trade of the Economic Belt and the Maritime Silk Road. International Journal of Industrial Engineering & Production Research, 33(3), 1-16.

-You G, Gan S, Guo H, Dagestani AA. Public Opinion Spread and Guidance Strategy under COVID-19: A SIS Model Analysis. Axioms. 2022; 11(6):296. https://doi.org/10.3390/axioms11060296

-A. A. Dagestani and L. Qing, "The Impact of Environmental Information Disclosure on Chinese Firms' Environmental and Economic Performance in the 21st Century: A Systematic Review," in IEEE Engineering Management Review, 2022, doi: 10.1109/EMR.2022.3210465.

-Zhao, S., Tian, W., & Dagestani, A. A. (2022). How do R&D factors affect total factor productivity: based on stochastic frontier analysis method. Economic Analysis Letters, 1(2), 28-34. https://doi.org/10.58567/eal01020005

-Bin He, Xiang Ma, Muhammad Nasir Malik, Riazullah Shinwari, Yaode Wang, Lingli Qing, Abd Alwahed Dagestani & Mohammed Moosa Ageli (2022) Sustainable economic performance and transition towards cleaner energy to mitigate climate change risk: evidence from top emerging economies, Economic Research-Ekonomska Istraživanja, DOI: 10.1080/1331677X.2022.2154240

-Shen, B., Yang, X., Xu, Y. et al. Can carbon emission trading pilot policy drive industrial structure low-carbon restructuring: new evidence from China. Environ Sci Pollut Res (2023). https://doi.org/10.1007/s11356-023-25169-4

-the manuscript still have a point not very clear , I hope they may explain ,what's your contribution to the theory ?you can choose the theory you think it fits better this article can be a good example could help you to understand what do I mean by "contribution to the theory" Eesley, C., Li, J. B., & Yang, D. (2016). Does institutional change in universities influence high-tech entrepreneurship? Evidence from China's Project 985. Organization Science, 27(2), 446-461. https://doi.org/10.1287/orsc.2015.1038

-It would be better if the authors used visual methods in the article to assist in presenting the problem or results.

-The correspondence between the conclusion and implication is not clear enough. Please polish them.

-please use the most new data

regards, go ahead

Reviewer #2: The authors developed a model to compute the optimal trade-off solutions for minimizing the risk of infection and maximizing the productivity in companies during the COVID-19 pandemic. They proposed a probabilistic analysis approach to compute the expected number of infected employees over the time by incorporating basic influential parameters such as the local incidence level, number of contacts among the employees and their average test interval and vaccination rate. The paper is good but needs some improvements. In the introduction, it should focus more on the objective of this work and the importance of it, and why it is essential for its contribution to the literature.

More updated references (year 2021, 2021, 2022, 2033) should be added to the literature section.

Recommendations should also be made in the last section of the text. For companies, adopting a Sustainable Development Goals (SDGs) approach is very important in the aftermath of the Covid-19 pandemic. The importance of the SDGs in business should be mentioned as a major future implication. Here are some references you can cite and include:

Cifuentes-Faura, J. (2022). Circular economy and sustainability as a basis for economic recovery post-COVID-19. Circular Economy and Sustainability, 2(1), 1-7.

Van Zanten, J. A., & Van Tulder, R. (2020). Beyond COVID-19: Applying “SDG logics” for resilient transformations. Journal of International Business Policy, 3(4), 451-464.

Cifuentes-Faura, J. (2021). COVID-19 and the opportunity to create a sustainable world through economic and political decisions. World Journal of Science, Technology and Sustainable Development, 18(4), 417-421.

Mattera, M., Gonzalez, F. S., Ruiz-Morales, C. A., & Gava, L. (2021). Facing a global crisis-how sustainable business models helped firms overcome COVID. Corporate Governance: The International Journal of Business in Society.

Reviewer #3: This study presented a COVID-19 inspired probabilistic model to propose the optimal solutions for employees' workplace presence during pandemics necessary for employee productivity. The topic is interesting and novel, and the manuscript is well written; however, some minor modifications must be incorporated before it is accepted for publishing. I have the following comments for the author(s) to consider;

1. Add the statistical results of the experiment performed in the abstract.

2. The introduction weakly documents the need and motivation for this study and designing the model.

3. The contributions of this work to the existing literature must be incorporated into the introduction section.

4. The author(s) should elaborate on the benefits of the designed model it can provide to an organization and compare it with previously designed models in more detail, which are discussed in the related work section.

5. Correct Figure and Fig. throughout the manuscript. The figure caption should be the same as used in the text (either Figure or Fig.) according to the guidelines of the Journal.

6. The implications of this study and model to the companies or organizations should be discussed in the conclusion.

7. Line 11. "number of employees presented" should be "number of employees presence".

8. Line 314 includes 11. The 11 should be either Eq.(11) or as defined by the author(s).

9. Line 435 in the conclusion section includes "First". I do not find any "second" or "third" in conclusion.

6. PLOS authors have the option to publish the peer review history of their article (what does this mean?). If published, this will include your full peer review and any attached files.

Reviewer #1: No

Reviewer #2: No

Reviewer #3: No

---

## [Author Response · Author response to Decision Letter 0]

12 Apr 2023

We uploaded 3 separated "Response to Reviewer" files, as well as one file as the "Letter to the Editor".

---

## [Decision Letter · Decision Letter 1]

24 Apr 2023

PONE-D-23-00247R1On the optimal presence strategies for workplace during pandemics: A COVID-19 inspired probabilistic modelPLOS ONE

Dear Dr. DavoodiMonfared,

Thank you for submitting your manuscript to PLOS ONE. After careful consideration, we feel that it has merit but does not fully meet PLOS ONE’s publication criteria as it currently stands. Therefore, we invite you to submit a revised version of the manuscript that addresses the points raised during the review process.

We look forward to receiving your revised manuscript.

Kind regards,

Muhammad Farhan Bashir

Academic Editor

PLOS ONE

Journal Requirements:

Reviewers' comments:

Reviewer's Responses to Questions

**Comments to the Author**

1. If the authors have adequately addressed your comments raised in a previous round of review and you feel that this manuscript is now acceptable for publication, you may indicate that here to bypass the “Comments to the Author” section, enter your conflict of interest statement in the “Confidential to Editor” section, and submit your "Accept" recommendation.

Reviewer #1: All comments have been addressed

Reviewer #2: All comments have been addressed

2. Is the manuscript technically sound, and do the data support the conclusions?

Reviewer #1: Yes

Reviewer #2: Yes

3. Has the statistical analysis been performed appropriately and rigorously? 

Reviewer #1: N/A

Reviewer #2: Yes

4. Have the authors made all data underlying the findings in their manuscript fully available?

Reviewer #1: Yes

Reviewer #2: Yes

5. Is the manuscript presented in an intelligible fashion and written in standard English?

Reviewer #1: Yes

Reviewer #2: Yes

6. Review Comments to the Author

Reviewer #1: well done

Reviewer #2: The manuscript has improved a lot. I only recommend that authors improve the quality of figures before publication.

7. PLOS authors have the option to publish the peer review history of their article (what does this mean?). If published, this will include your full peer review and any attached files.

Reviewer #1: **Yes: **Abd Alwahed Dagestani

Reviewer #2: No

---

## [Author Response · Author response to Decision Letter 1]

26 Apr 2023

Please see the response file, Response to Reviewers_JMC.pdf.

---

## [Editor Report · Decision Letter 2]

27 Apr 2023

On the optimal presence strategies for workplace during pandemics: A COVID-19 inspired probabilistic model

PONE-D-23-00247R2

Dear Dr. DavoodiMonfared,

We’re pleased to inform you that your manuscript has been judged scientifically suitable for publication and will be formally accepted for publication once it meets all outstanding technical requirements.

Kind regards,

Muhammad Farhan Bashir

Academic Editor

PLOS ONE
---

## [Editor Report · Acceptance letter]

3 May 2023

PONE-D-23-00247R2 

On the optimal presence strategies for workplace during pandemics: A COVID-19 inspired probabilistic model 

Dear Dr. Davoodi:

I'm pleased to inform you that your manuscript has been deemed suitable for publication in PLOS ONE. Congratulations! Your manuscript is now with our production department. 

Kind regards, 

on behalf of

Dr Muhammad Farhan Bashir 

Academic Editor

PLOS ONE